# Efficient Attenuation of Dextran Sulfate Sodium-Induced Colitis by Oral Administration of 5,6-Dihydroxy-8Z,11Z,14Z,17Z-eicosatetraenoic Acid in Mice

**DOI:** 10.3390/ijms22179295

**Published:** 2021-08-27

**Authors:** Shinya Takenouchi, Daiki Imai, Tatsuro Nakamura, Takahisa Murata

**Affiliations:** Department of Animal Radiology, Graduate School of Agricultural and Life Sciences, The University of Tokyo, Tokyo 113-8657, Japan; shinya-takenouchi88@g.ecc.u-tokyo.ac.jp (S.T.); d.imai8945@hotmail.com (D.I.); anakamu@g.ecc.u-tokyo.ac.jp (T.N.)

**Keywords:** 5,6-DiHETE, lipid mediator, pharmacokinetics, colitis, mouse model

## Abstract

5,6-dihydroxy-8Z,11Z,14Z,17Z-eicosatetraenoic acid (5,6-DiHETE) is an eicosapentaenoic acid-derived newly discovered bioactive anti-inflammatory lipid mediator having diverse functions. Here, we assessed the potential of orally administered 5,6-DiHETE in promoting healing of dextran sulfate sodium (DSS)-induced colitis in mice. We measured the plasma concentrations of 5,6-DiHETE in untreated mice before and 0.5, 1, 3, and 6 h after its oral administration (150 or 600 μg/kg) in mice. Mice developed colitis by DSS (2% in drinking water for 4 days), and 5,6-DiHETE (150 or 600 μg/kg/day) was orally administered from day 9 to 14. Next, the faecal hardness and bleeding were assessed, and the dissected colons on day 14 via H&E staining. The plasma concentration of 5,6-DiHETE reached 25.05 or 44.79 ng/mL 0.5 h after the administration of 150 or 600 μg/kg, respectively, followed by a gradual decrease. The half-life of 5,6-DiHETE was estimated to be 1.25–1.63 h. Diarrhoea deteriorated after day 3 and peaked on day 5, followed by a gradual recovery. Histological assessment on day 14 showed DSS-mediated granulocyte infiltration, mucosal erosion, submucosal edema, and cryptal abscesses in mice. Oral administration of 150 or 600 μg/kg/day of 5,6-DiHETE accelerated the recovery from the DSS-induced diarrhoea and significantly ameliorated colon inflammation. The therapeutic effect of 600 μg/kg/day 5,6-DiHETE was slightly stronger than that by 150 μg/kg/day. Our study reveals attenuation of DSS-induced colitis in mice by the oral administration of 5,6-DiHETE dose-dependently, thereby suggesting a therapeutic potential of 5,6-DiHETE for inflammatory bowel disease.

## 1. Introduction

Inflammatory bowel disease (IBD), characterized by repeated remissions and relapses, comprises two forms of chronic gut inflammation; ulcerative colitis (UC) and Crohn’s disease [1]. IBD causes severe intestinal and extra-intestinal symptoms that substantially worsen patients’ quality of life [2]. Since the global prevalence of IBD patients sharply increased from 3.7 million in 1990 to 6.8 million in 2017 [3], the need of the hour is to investigate potential therapeutic strategies to try and combat the disease.

Currently, 5-aminosalicylic acid, or corticosteroids, and anti-tumour necrosis factor-α antibodies are the major modes of treatment for IBD. Some of these medications that attenuate IBD symptoms to some extent have been reported to exert substantial adverse effects such as nasopharyngitis, hypokalemia, and leukopenia [4,5,6]. Several studies have suggested the beneficial potential of other drugs attenuating symptoms of animal colitis models [7,8]. In addition, there are several reports proposing new alternatives such as nutritional diet therapies with fewer side effects. For example, curcumin, a natural antioxidant substance from turmeric, has been reported to be therapeutically effective against mice colitis and UC in patients [9,10]. Also, it has been shown that treatment with ω-3 polyunsaturated fatty acids (PUFAs) such as docosahexaenoic acid (DHA) and eicosapentaenoic acid (EPA) and fish oil lead to attenuation of intestinal inflammation in UC patients and rodents with colitis [11,12,13]. However, only a considerably high dosage of the fatty acids (dozens to hundreds mg/kg) showed beneficial effects [11,12,13]. Additionally, others have reported insignificant or even negative results of ω-3 PUFAs treatment against colitis. A double-blind, randomized, and placebo-controlled trial with 63 UC patients in the UK showed that ω-3 PUFAs combining diet led to a slight increase in the relapse rates during the 12-month test period [14]. Along similar lines, Woodworth et al. reported that a DHA-rich fish oil diet exacerbated *Helicobacter hepaticus*-induced murine colitis [15]. Such observations create the need for a careful and detailed investigation of the ω-3 PUFAs’ mechanism of action for future applications.

It is known that PUFAs undergo enzymatic or non-enzymatic metabolization in our bodies. PUFAs are metabolized by the enzymes cyclooxygenase, lipoxygenase, and cytochrome p450 (CYP) into various metabolites representing pro-inflammatory or anti-inflammatory reactions. 5,6-dihydroxy-8Z,11Z,14Z,17Z-eicosatetraenoic acid (5,6-DiHETE) is a CYP metabolite from EPA [16]. Our previous studies show that the level of 5,6-DiHETE increases in colon tissue during the healing stage of dextran sulfate sodium (DSS)-induced colitis in mice [17,18]. We have also shown the inhibition of histamine-induced vascular hyperpermeability by antagonizing the transient receptor potential vanilloid (TRPV) 4 channels upon pre-treatment with 5,6-DiHETE (0.1 μg, intracutaneously) [18,19]. Furthermore, intraperitoneal injection of 5,6-DiHETE (50 μg/kg every second day) promoted the recovery from DSS-induced colitis in mice [18]. These results suggest that the EPA-metabolite 5,6-DiHETE can be a potential new alternative to nutritional supplements to treat colitis. However, the mode of action and the appropriate dose for orally administered 5,6-DiHETE in treating colitis needs to be investigated further.

In this study, we used a DSS-induced colitis murine model to assess the potential of orally administered 5,6-DiHETE for treating colitis.

## 2. Results

### 2.1. Pharmacokinetics of 5,6-DiHETE in Mice Plasma

We first investigated the pharmacokinetics of 5,6-DiHETE in the plasma of mice. We orally administered 5,6-DiHETE (150 or 600 μg/kg) or vehicle to mice and measured the plasma concentration of 5,6-DiHETE with liquid chromatography-tandem mass spectrometry (LC/MS) at intended time-points (before and 0.5, 1, 3, and 6 h after the administration, Figure 1). We determined the doses of 5,6-DiHETE based on our previous research [18]. In the vehicle-administered mice, the plasma level of 5,6-DiHETE was consistently less than 1.00 ng/mL at all the measured time-points. Consistently, before the administration, the plasma levels of 5,6-DiHETE were also low similar to the vehicle-treated group. In contrast, in 5,6-DiHETE-treated mice, its concentrations were the highest 0.5 h after the administration throughout the experimental period (150 μg/kg; 25.05 ± 7.20 ng/mL and 600 μg/kg; 44.79 ± 7.62 ng/mL, respectively), followed by a gradual decrease, thereby suggesting that the elimination of 5,6-DiHETE already started 0.5 h after the administration. Our investigations also revealed that 5,6-DiHETE (above 15% of 0.5 h) remained partially in the plasma 3 h after the administration, which led us to estimate the half-lives of 150 and 600 μg/kg 5,6-DiHETE to be 1.25 and 1.63 h, respectively.

### 2.2. Oral Administration of 5,6-DiHETE Did Not Affect Body Weight Change but Improved Faecal Condition

We next assessed the effects of orally administered 5,6-DiHETE (150 or 600 μg/kg/day, from day 9 to 14) to DSS-induced colitis mice. We measured the body weight and monitored the faecal condition of mice every day (Figure 2a,b, respectively). The body weight of naive (untreated) mice gradually increased, while that of DSS-treated mice began to drop from day 4 or 5 to reach a minimum around day 7 and gradually recovered to the initial level by day 14. We observed that administration of neither 150 nor 600 μg/kg/day of 5,6-DiHETE from day 9 influenced the recovery of the body weight of the mice compared with that of the vehicle group (Figure 2a).

Additionally, faecal scores (ranging 0 to 5) were determined based on the stool condition varying from normal hard to wet, soft, and watery faeces, including bloody diarrhoea. The naive mice showed faecal scores below 1 throughout the experimental period. In contrast, the scores in all the DSS-treated groups started to elevate above 1 from day 2 or 3 to reach a maximum score of 4.3 ± 0.2 on day 5 before gradually decreasing to about 2. Moreover, the administration of 150 or 600 μg/kg/day of 5,6-DiHETE significantly accelerated the decrease of faecal scores from day 9 compared with vehicle administration. On day 14, the faeces of vehicle-treated colitis mice were still wet and soft, whereas in 5,6-DiHETE-treated mice, it regained its normal appearance, and the faecal scores were significantly suppressed (Figure 2b, DSS + Vehicle; 2.1 ± 0.3, DSS + 5,6-DiHETE 150 μg/kg/day; 0.8 ± 0.3, DSS + 5,6-DiHETE 600 μg/kg/day; 0.8 ± 0.4). These results collectively indicated the potential of 5,6-DiHETE in selectively improving only the faecal condition and not the body weight loss in colitis recovery.

### 2.3. 5,6-DiHETE Administration Attenuated DSS-Induced Colon Inflammation in Mice

The histological assessment of colon tissues obtained on day 14 from naive mice showed normal intestinal structures (a rare infiltration of granulocytes and intact mucosal epithelia) without any inflammatory sign (Figure 3a, double-headed arrow indicating the width of submucosa). However, in vehicle-treated DSS colitis mice on day 14, the colons showed dilated submucosa/mucosa (double-headed arrow), indicating tissue edemas. Also, it exhibited severe epithelial damage leading to frequent erosions (star shape) or ulcerations and a heavy infiltration of granulocytes (arrowheads) related to multifocal abscesses (areas surrounded by dotted line) in the crypts and mucosa (Figure 3b). On the other hand, oral administration of 150 μg/kg/day of 5,6-DiHETE suppressed submucosal/mucosal edema (double-headed arrow), thereby preventing the DSS-induced collapse in epithelial structure. The lower dose of 5,6-DiHETE mediated a partial attenuation of granulocytes infiltration (arrowheads) and formation of cryptal abscesses (areas surrounded by dotted line) compared with the vehicle-treated colon (Figure 3c). Moreover, the administration of 600 μg/kg/day 5,6-DiHETE substantially ameliorated edema (double-headed arrow), tissue damage, and decreased the number of infiltrating cells (arrowheads) similar to the action of 150 μg/kg/day dose. In addition, 600 μg/kg/day 5,6-DiHETE abolished the abscess in crypts and mucosa (Figure 3d). The numbers of infiltrating granulocytes counted per high-power field were shown in Figure 3e. There was a drastic increase in vehicle-treated DSS colitis mice compared to naive (1.1 ± 0.2 and 11.0 ± 1.1, respectively). In contrast, oral administration of 150 or 600 μg/kg/day of 5,6-DiHETE significantly suppressed the numbers of infiltrating granulocytes (4.5 ± 0.5 or 3.5 ± 0.2, respectively).

The results of histological scoring in each index are shown in Table 1. Compared to naive mice, the vehicle-treated DSS colitis mice exhibited significantly higher scores in total (1.2 ± 0.5 and 9.4 ± 0.6, respectively) and in each index, indicating a higher degree of damage. The DSS colitis mice administered 150 or 600 μg/kg/day of 5,6-DiHETE showed significantly lower total scores than the vehicle-treated DSS colitis mice (5.9 ± 0.9 or 5.1 ± 0.6, respectively). In addition, compared to vehicle-treated DSS colitis mice, 150 μg/kg/day of 5,6-DiHETE treatment of the mice showed a slight decrease in the scores of cryptitis (1.8 ± 0.1 and 1.3 ± 0.3, respectively), whereas 600 μg/kg/day showed a significant decrease (0.6 ± 0.3). These data corroborated our observations that much fewer cryptal abscesses were seen in 600 μg/kg/day of 5,6-DiHETE administration than in the group receiving 150 μg/kg/day.

Collectively from our data, we conclude that the oral administration of 5,6-DiHETE has the potential to accelerate the recovery from DSS-induced colitis.

## 3. Discussion

In the present study, we initially confirmed the absorbance and persistence of 5,6-DiHETE in mouse plasma over time upon oral intake. Next, we demonstrated its therapeutic effects on DSS-induced murine colitis by showing that 5,6-DiHETE administration significantly inhibited DSS-induced diarrhoea, granulocytes infiltration, erosion, edema, and abscess formation in the colon. The effects of oral administration of 5,6-DiHETE against colitis indicated a therapeutic potential with pharmacokinetic evidence for future use.

The burdens, including adverse effects, necessity of continuous treatment, and considerable expenses, created on patients by the currently available symptomatic therapies for IBD creates a need to identify new alternatives such as nutritional diet therapies. ω-3 PUFAs and fish oil have been focused as alternative options for IBD treatment. However, there were some reports showing its ineffectiveness against IBD and adverse effects [20,21]. And even in case where its beneficial effects were shown, a relatively large intakes of such fatty acids were required. Cho et al. reported that concurrent oral administration of 30 mg/kg/day of DHA for 7 days could ameliorate colon inflammation and cryptal destruction in DSS-induced colitis in mice [12]. Morin et al. also showed that concurrent oral administration of 318 mg/kg/day of EPA monoglyceride for 12 days prevented diarrhoea, leucocytes infiltration, and mucin production in DSS-induced colitis rats [13]. These reports showed how a prolonged treatment of fatty acids is required for any effective treatment of colitis. In contrast, our study demonstrates how the oral administration of EPA-metabolite 5,6-DiHETE attenuates the DSS-induced colitis at a much lower dose than the previous studies; 150 or 600 μg/kg/day over a shorter period (6 days). The distinct actions between ω-3 PUFAs and 5,6-DiHETE could be attributed to the differences in their mechanisms of action. The former exerts beneficial effects by competitive inhibition of enzymes common to the arachidonic acid pathway and the metabolism into anti-inflammatory mediators like resolvins and protectins [22]. In contrast, the anti-inflammatory action of 5,6-DiHETE directly blocks a polymodal non-selective calcium channel TRPV4 [18], which presumably leads to faster and more potent therapeutic effects against colitis.

Studies have shown that the activation of TRPV4 by intracolonic administration of agonists in mice acutely induces pro-inflammatory cytokine releases and colon inflammation [23], whereas pharmacological TRPV4 antagonism abolishes the 2,4,6-trinitrobenzenesulfonic acid (TNBS)-induced colon inflammation [24]. Furthermore, it is known that TRPV4 deficiency attenuates colon inflammation by suppressing vascular hyperpermeability in DSS-induced colitis mice [25]. This study highlights how the oral supplementation of 5,6-DiHETE can be beneficial against various types of colitis. However, a limitation of our study is the difference in the pathology of colitis between animal models and human patients. Despite the increased TRPV4 expressions in the colon of both human IBD patients and animal colitis models [23,24,25], the extent of TRPV4 activation to drive human IBD remains largely unexplored. Hence, further trials are needed to assess the effectiveness of 5,6-DiHETE in the treatment of human colitis.

Previous double-blind and randomized studies have highlighted that the use of a high dosage of ω-3 diet (1 g/day of ω-3 diet containing 460 mg EPA and 380 mg DHA) inhibits thrombin formation and fibrin clot aggregation in patients with coronary artery disease [26]. Moreover, such a high intake of ω-3 fatty acids is assumed to disturb thromboxane synthesis necessary for platelet aggregation [27]. Therefore, concomitant administration of ω-3 fatty acids with anticoagulation and antiplatelet drugs might lead to adverse effects. In this study, we hypothesized that 5,6-DiHETE, which does not influence the synthesis of other lipid metabolites, can be a safer nutritional supplement with fewer adverse effects.

We estimated the half-life of 5,6-DiHETE to be 1.25–1.63 h. Pharmacokinetics of lipids vary with many factors, such as liver enzymatic profile, renal clearance function, blood stream, and tissue distribution. Furthermore, the pharmacological half-life of 5,6-DiHETE in mice is likely shorter by one or two orders of magnitude than in humans [28]. However, further investigations are required to reveal the exact kinetic profile of 5,6-DiHETE in humans.

We used DSS-induced colitis model which is supposed to accurately mimic the acute colitis in human. DSS causes intestinal epithelial barrier disruption in colon and then entry of luminal bacteria or bacteria-related antigens into mucosa. This leads to erosions or ulcerations and production of proinflammatory cytokines or chemokines which results in infiltration of innate followed by adoptive immune cells [29]. Such pathologic mechanism is very similar to human IBD except that chronic DSS colitis is driven by mixed reaction of innate cells and T cells whereas human chronic IBD is more inclined to T cells-dependent immunity [30]. Since DSS colitis model presumably mimics the innate immune cells-driven acute colitis, 5,6-DiHETE might behave differently in chronic human IBD. Unlike DSS model, TNBS-induced colitis model is characterized by T cells-driven colonic inflammation [31]. As mentioned above, Fichna et al. demonstrated the beneficial effects of a selective TRPV4 antagonist on the TNBS-induced murine colitis [24]. Thus, 5,6-DiHETE can be beneficial for various types of colitis.

Our previous study revealed that the intestines of blue back fishes, especially those of sardines, were rich in 5,6-DiHETE (970.22 ± 221.24 ng/g) [32]. Fish oil and EPA supplements are sometimes poorly accepted by patients because of their unpleasant taste and smell or side effects, including eructation, nausea, fishy odor, and abdominal bloating [33,34,35]. Hence, one of the primary objectives of our study is to establish an efficient extraction procedure for 5,6-DiHETE, which will lead to a safer and more tolerable nutrient for the patients. Further investigations are needed to establish an extraction method and to investigate metabolism and adverse effects of 5,6-DiHETE.

In contrast to the present study, our previous study showed that intraperitoneally injected 50 μg/kg/day of 5,6-DiHETE accelerated the recovery from diarrhoea, colon inflammation, and body weight loss in DSS-induced colitis in mice [18]. This difference in the mechanism of action of 5,6-DiHETE can be attributed to body weight changes influenced by many factors, including food and water consumption, temperature, humidity, and stress. The inconsistency between our two reports may be due to these differences, with the amount of 5,6-DiHETE absorbed via the intragastric route not reaching enough levels to affect body weight change.

Collectively, here we highlight that orally administered 5,6-DiHETE, a newly discovered anti-inflammatory lipid mediator, could exert therapeutic efficacy in the DSS-induced colitis murine model. Although further clinical and experimental studies need to clarify the substantial role and mode of action of 5,6-DiHETE, this finding implies a potential therapeutic use of 5,6-DiHETE as nutritional supplements for colitis treatment.

## 4. Materials and Methods

### 4.1. Animals and Reagents

All animal experiments were approved by the Institutional Animal Care and Use Committee of the University of Tokyo (approval number: P16-209). Male 6 to 8-week-old C57BL/6 mice were used in experiments. Mice were kept in a room on a 12 h/12 h light/dark cycle. Average room temperature was regulated in the range from 20 to 25 °C, average humidity 40–60%. Mice were able to freely access food and water ad libitum during the experiments.

DSS was purchased from MP Bio Japan K.K. (Tokyo, Japan). 5,6-DiHETE and 14,15-dihydroxy-5Z,8Z,11Z-eicosatrienoic-16,16,17,17,18,18,19,19,20,20,20-d_11_ acid (14,15-DiHET-d_11_) were purchased from Cayman Chemical (Ann Arbor, MI, USA).

### 4.2. Pharmacokinetic Study

5,6-DiHETE was orally administered to mice (n = 8 per group) at 150 or 600 μg/kg, doses determined based on our previous study [18]. Phosphate buffered saline (PBS; 8.1 mM Na_2_HPO_4_, 1.5 mM KH_2_PO_4_, 1.4 M NaCl and 2.7 mM KCl, pH 7.4) was administrated as vehicle control. Blood samples were collected before and after the dosing (0.5, 1, 3 and 6 h after the administration). Separated plasma samples were stored at −80 °C until measurement.

The average of plasma concentrations of 5,6-DiHETE was plotted against time. The elimination rate constant (*K_el_*) was determined by the slope of the linear regression line for the semi-log plot of concentration versus time curve. The biological half-life of 5,6-DiHETE was calculated using the formula.
$$\left(\mathrm{Half}-\mathrm{life}\right)=\frac{ln\left(2\right)}{K_{el}}$$

### 4.3. Measurement of 5,6-DiHETE with LC/MS

Internal standard, 14,15-DiHET-d_11_ (10 ng/mL as final concentration), was added to the stored plasma. This solution was deproteinized by mixing with organic solvent (methanol: acetonitrile = 1:1, *v*/*v*, containing 5N HCl as final concentration). After centrifugation (20,000 *g*, 10 min, 4 °C), distilled water was added to the supernatants and this mix was loaded to Oasis PRiME HLB cartridges (Waters, Milford, MA, USA). After washing by distilled water and hexane, lipids were eluted by 50 μL methanol. The eluted sample solutions were filtered with Ultrafree-MC Centrifugal Filter (Merck KGaA, Darmstadt, Germany). The collected analytes were used for measurement.

The analytes were introduced into an LC/MS-8060 system (Shimadzu, Kyoto, Japan). The liquid chromatographic separation was conducted using a Kinetex C8 column (Phenomenex, Torrance, CA, USA) using a mobile phase consisting of 0.1% (*v*/*v*) formic acid (solvent A) and acetonitrile (solvent B). The following gradient was used at a flow rate of 0.4 mL/min: initial at 10:90 (solvent A: solvent B), to 1 min at 50:50 and 1 to 5 min at 34:66. The LC/MS was operated in the negative ion mode. For analysis, the monitored transition for 5,6-DiHETE was *m*/*z* 335.20 > 145.10 and for 14,15-DiHET-d_11_ it was *m*/*z* 348.30 > 207.10. 5,6-DiHETE was identified by elution time: 4.103 min and 14,15-DiHET-d_11_ was 4.178 min.

### 4.4. Colitis Models and 5,6-DiHETE Administration

As shown in Figure 4, mice (n = 5 to 9 per group) were received 2% (*w*/*v*) of DSS (molecular weight 36,000–50,000) in drinking water for 4 consecutive days and then it was replaced with water. Naive mice were always received water ad libitum. DSS-treated mice were randomly grouped and 5,6-DiHETE (150 or 600 μg/kg/day) or vehicle (PBS) was orally administered from day 9 to 14. Faecal condition and body weight were monitored every day. Mice were euthanized on day 14 and the colons were dissected for histological assessments.

### 4.5. Evaluation of Faecal Condition

The faecal condition score was determined on stool consistency and bleeding in faeces. Each score was defined as follows: stool consistency (0 for normal; 1 for soft but still formed; 2 for very soft; 3 for watery diarrhoea) and bleeding (0 for normal; 1 for occult blood; 2 for gross bleeding). Bleeding was detected using hemoccult single slides (Beckman Coulter, Inc., Brea, CA, USA). Faecal scores were calculated by adding these two values. The total score ranges from 0 to 5.

### 4.6. Histological Assessment

Dissected colons were washed with PBS, fixed in 4% (*w*/*v*) paraformaldehyde, and embedded in paraffin for histological analysis. Slides were sectioned at 4 μm thick and stained with hematoxylin and eosin following conventional methods. Histological scoring was performed based on a previous study [36]. The detailed indices of histological scoring are shown in Table 2.

### 4.7. Statistical Analyses

The results of the experiments were expressed as mean ± standard error of the mean (SEM). Statistical evaluation for comparison between two groups was performed by unpaired Student’s *t*-test for the parametric data and Mann-Whitney U test for non-parametric data. Comparison between multiple groups was performed using one-way ANOVA followed by Tukey’s test for parametric data and Kruskal-Wallis test followed by Steel-Dwass test for non-parametric data. A value of *p* < 0.05 was considered statistically significant.

## Figures and Tables

**Figure 1 ijms-22-09295-f001:**
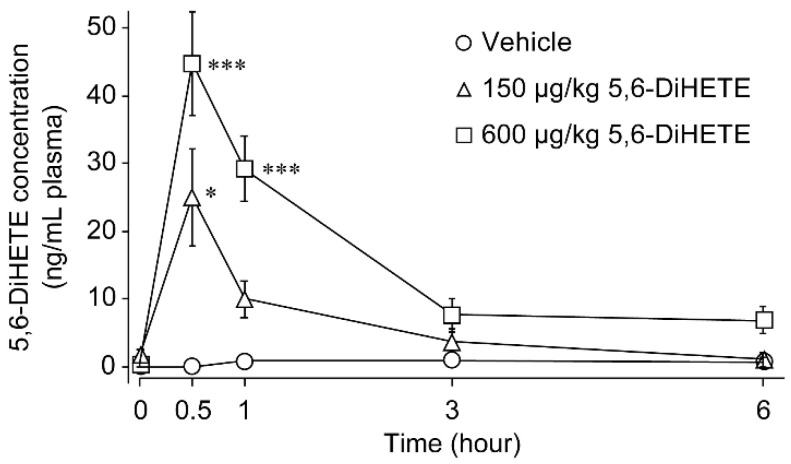
Pharmacokinetics of 5,6-DiHETE in plasma after oral administration. Plasma samples were collected before and 0.5, 1, 3 and 6 h after dosing 150 or 600 μg/kg of 5,6-DiHETE or vehicle (n = 8 per group). The concentration of 5,6-DiHETE was measured with LC/MS. Data are represented as mean ± SEM. * *p* < 0.05, *** *p* < 0.001 compared with Vehicle.

**Figure 2 ijms-22-09295-f002:**
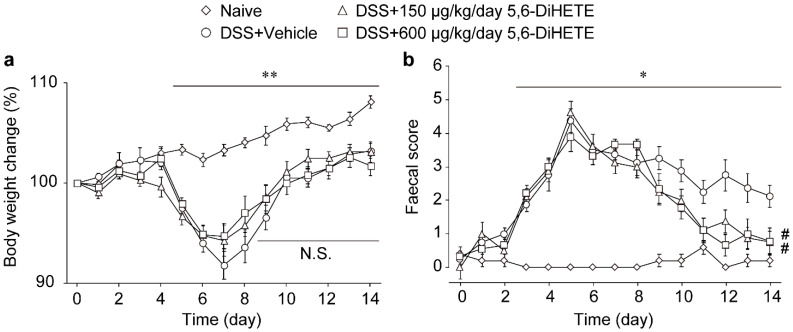
Effects of oral administration of 5,6-DiHETE on the body weight change or faecal score. (**a**) Body weight and (**b**) faecal scores were assessed as pathological markers (n = 5–9). Body weight was normalized by that of day 0. Data are represented as mean ± SEM. * *p* < 0.05, ** *p* < 0.01 between Naive and DSS + Vehicle. ^#^ *p* < 0.05, compared with DSS + Vehicle. N.S., not significant between DSS + Vehicle, DSS + 5,6-DiHETE (150 μg/kg/day) and DSS + 5,6-DiHETE (600 μg/kg/day). Naive, non-treated healthy mice; DSS + Vehicle, vehicle-treated DSS colitis mice; DSS + 5,6-DiHETE (150 μg/kg/day), 150 μg/kg/day of 5,6-DiHETE-treated DSS colitis mice; DSS + 5,6-DiHETE (600 μg/kg/day), 600 μg/kg/day of 5,6-DiHETE-treated DSS colitis mice.

**Figure 3 ijms-22-09295-f003:**
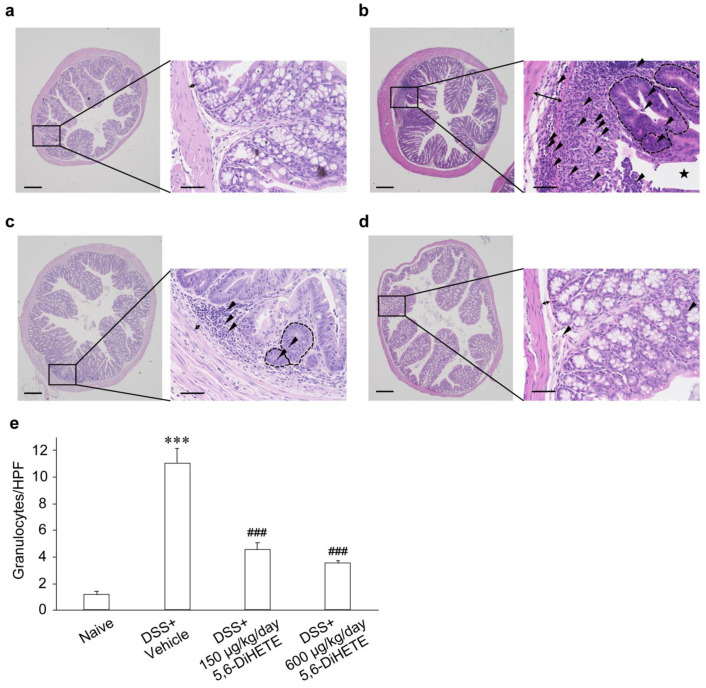
Effects of oral administration of 5,6-DiHETE on inflammation in colon on day 14. The sections of dissected colons on day14 were stained with hematoxylin and eosin. Representative images of (**a**) Naive (n = 5), (**b**) DSS + Vehicle (n = 8), (**c**) DSS + 5,6-DiHETE (150 μg/kg/day, n = 8) and (**d**) DSS + 5,6-DiHETE (600 μg/kg/day, n = 9) in low-power field (left panel, magnification 40×, scale bar = 250 μm) and high-power field (HPF, right panel, magnification 400×, scale bar = 50 μm). Double-headed arrows indicate the width of submucosa. Arrowheads indicate infiltrated granulocytes. Areas surrounded by dotted line indicate cryptal abscesses. Star shape indicates mucosal erosion. (**e**) The number of granulocytes in HPF (magnification 400×). Granulocytes were counted in randomly selected 25 fields for each tissue. Then, the mean value of them was adopted as the number of granulocytes in the tissue. Data are represented as mean ± SEM. *** *p* < 0.001 compared with Naive. ^###^ *p* < 0.01 compared with DSS + Vehicle. Naive, non-treated healthy mice; DSS + Vehicle, vehicle-treated DSS colitis mice; DSS + 5,6-DiHETE (150 μg/kg/day), 150 μg/kg/day of 5,6-DiHETE-treated DSS colitis mice; DSS + 5,6-DiHETE (600 μg/kg/day), 600 μg/kg/day of 5,6-DiHETE-treated DSS colitis mice.

**Figure 4 ijms-22-09295-f004:**
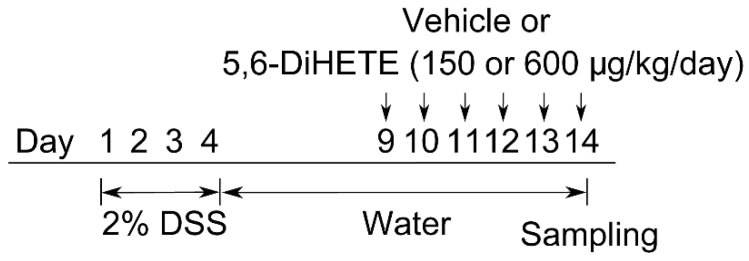
Schematic of experimental design. Colitis was induced by 4 days administration of DSS (2%, *w*/*v*) in drinking water. These mice received normal water from day 5. From day 9 to 14, we orally administered DSS-induced colitis mice 150 μg/kg/day (n = 8) or 600 μg/kg/day (n = 9) of 5,6-DiHETE or vehicle (n = 8). Naive mice were fed with normal water and received no treatment throughout the experiment (n = 5). Body weight and faecal condition were assessed every day. On day 14, mice colons were dissected for histological assessment.

**Table 1 ijms-22-09295-t001:** Results of histological scoring in each of indices (Mucosal integrity, Cell infiltration, Cryptitis and Edema) and in total.

Groups	Mucosal Integrity (0~6)	Cell Infiltration (0~3)	Cryptitis (0~2)	Edema (0~1)	Total (0~12)
Naive	0.4 ± 0.4	0.4 ± 0.2	0.0 ± 0.0	0.4 ± 0.2	1.2 ± 0.5
DSS + Vehicle	4.0 ± 0.5 **	2.6 ± 0.2 **	1.8 ± 0.1 ***	0.8 ± 0.1	9.4 ± 0.6 ***
DSS + 5,6-DiHETE (150 μg/kg/day)	3.0 ± 0.6	1.1 ± 0.1 ^##^	1.3 ± 0.3	0.5 ± 0.2	5.9 ± 0.9 ^#^
DSS + 5,6-DiHETE (600 μg/kg/day)	3.1 ± 0.3	1.0 ± 0.0 ^##^	0.6 ± 0.3 ^#^	0.4 ± 0.2	5.1 ± 0.6 ^#^

Data are represented as mean ± SEM. ** *p* < 0.01, *** *p* < 0.001 compared with Naive. ^#^ *p* < 0.05, ^##^ *p* < 0.01 compared with DSS + Vehicle. Naive, non-treated healthy mice; DSS + Vehicle, vehicle-treated DSS colitis mice; DSS + 5,6-DiHETE (150 μg/kg/day), 150 μg/kg/day of 5,6-DiHETE-treated DSS colitis mice; DSS + 5,6-DiHETE (600 μg/kg/day), 600 μg/kg/day of 5,6-DiHETE-treated DSS colitis mice.

**Table 2 ijms-22-09295-t002:** Indices, criteria and scores for histological scoring.

Index	Criterion	Score
Mucosal integrity	Single cell death	2
	Erosions	4
	Florid ulcerations	6
Infiltration of granulocytes in the lamina propria mucosae	Number of cells at 400× magnification	
	=1~5	1
	=6~10	2
	>10	3
Cryptitis (granulocytes in cryptal epithelium)	Inflammatory cells in crypts	2
	Cryptal abscess	1
Mucosal/Submucosal edema	Existing	1
Total score		0~12

## Data Availability

The datasets generated and analysed during the study are available from the corresponding author on reasonable request.

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
