# Peer review of "Efficient Attenuation of Dextran Sulfate Sodium-Induced Colitis by Oral Administration of 5,6-Dihydroxy-8Z,11Z,14Z,17Z-eicosatetraenoic Acid in Mice"

_ijms, 2021, doi:10.3390/ijms22179295_

Round 1

Reviewer 1 Report

In this manuscript, Takenouchi et al reported the protective effect of 5,6-DiHETE in DSS-colitis model. Although their finding is of interest, most of the results in this manuscript were already shown in their previous FASEB paper. The authors are using the same model (DSS-colitis) and the same reagent (5,6-DiHETE), and carrying out the same analysis. The only difference is the dosage and the administration method of 5,6-DiHETE, and the measurement of the plasma 5,6-DiHETE concentration. It is recommended that the authors prove the efficacy of 5,6-DiHETE in another colitis model, such as TNBS colitis, T cell transfer model, etc., to convince the readers.   Minor point In P. 2 L. 79, the authors describe that the plasma level of 5,6-DiHETE before  5,6-DiHETE administration was especially low in vehicle-administered mice. Why is this? Shouldn't plasma level be equal between all mice? 

Reviewer 2 Report

The manuscript by Takenouch et al. Highlights the role of 5,6-DiHETE in DSS-induced colitis.

The manuscript is interesting and well organized.

I suggest the authors add this recent manuscript on the pharmacological action of 5,6-DiHETE.

 Kobayashi K, Ashina K, Derouiche S, Hamabata T, Nakamura T, Nagata N,Takenouchi S, Tominaga M, Murata T. 5,6-dihydroxy-8Z,11Z,14Z,17Z-eicosatetraenoic acid accelerates the healing of colitis by inhibiting transient receptor potential vanilloid 4-mediatedsignaling. FASEB J. 2021 Apr;35(4):e21238.

Also in the introduction I suggest to give some examples of other drugs that are used for colitis, please look at these manuscripts.

   Pagano E, Romano B, Iannotti FA, Parisi OA, D'Armiento M, Pignatiello S,Coretti L, Lucafò M, Venneri T, Stocco G, Lembo F, Orlando P, Capasso R, Di MarzoV, Izzo AA, Borrelli F. The non-euphoric phytocannabinoid cannabidivarincounteracts intestinal inflammation in mice and cytokine expression in biopsiesfrom UC pediatric patients. Pharmacol Res. 2019 Nov;149:104464. 

Sadeghi N, Mansoori A, Shayesteh A, Hashemi SJ. The effect of curcuminsupplementation on clinical outcomes and inflammatory markers in patients withulcerative colitis. Phytother Res. 2019 Dec 4. doi: 10.1002/ptr.6581.  

da Silva VC, de Araújo AA, de Souza Araújo DF, Souza Lima MCJ, Vasconcelos RC,de Araújo Júnior RF, Langasnner SMZ, de Freitas Fernandes Pedrosa M, de Medeiros CACX, Guerra GCB. Intestinal Anti-Inflammatory Activity of the Aqueous Extractfrom Ipomoea asarifolia in DNBS-Induced Colitis in Rats. Int J Mol Sci. 2018 Dec 12;19(12).

 Morgan S, Hooper KM, Milne EM, Farquharson C, Stevens C, Staines KA.Azathioprine Has a Deleterious Effect on the Bone Health of Mice with DSS-InducedInflammatory Bowel Disease. Int J Mol Sci. 2019 Dec 3;20(23). 

 Pagano E, Capasso R, Piscitelli F, Romano B, Parisi OA, Finizio S, Lauritano A, Marzo VD, Izzo AA, Borrelli F. An Orally Active Cannabis Extract with High Content in Cannabidiol attenuates Chemically-induced Intestinal Inflammation and  Hypermotility in the Mouse. Front Pharmacol. 2016 Oct 4;7:341. Capasso R, Orlando P, Pagano E, Aveta T, Buono L, Borrelli F, Di Marzo V, Izzo AA. Palmitoylethanolamide normalizes intestinal motility in a model of  post-inflammatory accelerated transit: involvement of CB₁ receptors and TRPV1 channels. Br J Pharmacol. 2014 Sep;171(17):4026-37.  Have the doses been chosen based on previous studies?

Do the authors think the microbiota can affect or is affected by 5,6-DiHETE?

In the Discussion, the Authors should highlight the possible clinical significance of their findings

Reviewer 3 Report

  • In the introduction part the authors stated that “… The multifactorial nature of IBD makes it difficult to specify its etiology, making symptomatic therapy the only treatment choice. Currently, 5-aminosalicylic acid, or corticosteroids, and anti-tumour necrosis factor-a antibodies are the major modes of treatment for IBD. These medications that attenuate IBD symptoms to some extent have been reported to exert substantial adverse effects such as nasopharyngitis, hypokalemia, and leucopenia…”. Nothing is more untrue. Today we have a plethora of drugs that have dramatically changed the natural history of patients with IBD. The authors should delete this paragraph. Instead, they should state differently the reasons that led them to carry out this experimental study.
  • The authors stated that “…ω-3 PUFAs and fish oil are already established as alternative options for IBD treatment…” However, as the authors described in other parts of the paper, a number of studies have consistently questioned the efficacy of ω-3 PUFAs, especially in clinical practice. So, there are reports showing that ω-3 PUFAs have no effect or even adverse effects on IBD. In randomized, placebo-controlled trials, ω-3 PUFAs intake had no effect in improving the recovery of colitis Turner et al Cochrane Database Syst. Rev.  Barbosa et al Nutrition. 2003;19:837–842 and has even enhanced disease activity in UC patients Dichi et al Nutrition. 2000;16:87–90. Treatment of fish oil has had little effect on DSS- or TNBS-induced colitis in rats Vieira de Barros et al Nutrition 2011;27:221–226. Shoda et al J Gastroenterol 1995;30(Suppl. 8):98–101, and has exacerbated the DSS-induced colitis in mice Matsunaga et al, Inflamm Bowel Dis 2008;14:1348–1357. They should modify their statement accordingly.
  • In my opinion, the authors should have a rather conservative attitude concerning the clinical application of their results. In the conclusion part they should emphasize the fact that further clinical and experimental studies are warranted to clarify the real role and mechanism(s) of action of the metabolites of ω-3 PUFAs.

Round 2

Reviewer 1 Report

The authors should consider of performing other colitis model in the future.